# Disentangled Contrastive Learning on Graphs

**Haoyang Li[1], Xin Wang[1]\*, Ziwei Zhang[1], Zehuan Yuan[2], Hang Li[2], Wenwu Zhu[1]\***
[1]Tsinghua University, [2]Bytedance
lihy18@mails.tsinghua.edu.cn, {xin_wang, zwzhang}@tsinghua.edu.cn,
{yuanzehuan, lihang.lh}@bytedance.com, wwzhu@tsinghua.edu.cn

## Abstract

Recently, self-supervised learning for graph neural networks (GNNs) has attracted considerable attention because of their notable successes in learning the representation of graph-structure data. However, the formation of a real-world graph typically arises from the highly complex interaction of many latent factors. The existing self-supervised learning methods for GNNs are inherently holistic and neglect the entanglement of the latent factors, resulting in the learned representations suboptimal for downstream tasks and difficult to be interpreted. Learning disentangled graph representations with self-supervised learning poses great challenges and remains largely ignored by the existing literature. In this paper, we introduce the Disentangled Graph Contrastive Learning (**DGCL**) method, which is able to learn disentangled graph-level representations with self-supervision. In particular, we first identify the latent factors of the input graph and derive its factorized representations. Each of the factorized representations describes a latent and disentangled aspect pertinent to a specific latent factor of the graph. Then we propose a novel factor-wise discrimination objective in a contrastive learning manner, which can force the factorized representations to independently reflect the expressive information from different latent factors. Extensive experiments on both synthetic and real-world datasets demonstrate the superiority of our method against several state-of-the-art baselines.

## 1 Introduction

Graph structured data is ubiquitous in the real world, e.g., social networks, biology networks, traffic networks, etc. Recently, graph neural networks (GNNs) have become increasingly prevalent in learning graph representations in a supervised manner, demonstrating their strength in a wide variety of research fields [1, 2, 3, 4]. GNNs require task-dependent annotated labels to learn effective representations, which are extremely scarce, or even unavailable in practice, thus motivating the advent of self-supervised graph representation learning.

Contrastive learning, as a discriminative approach pulling similar samples close and pushing dissimilar samples far away, has become a dominant strategy in self-supervised graph representation learning [5, 6, 7, 8, 9, 10, 11]. Despite their notable successes, the existing graph contrastive learning methods generally adopt a holistic scheme, i.e., the learned representations characterize graphs as a perceptual whole, ignoring the nuances between different aspects of the graph. In fact, the formation of a graph typically follows a relational process in the real world, driven by many complex latent factors. For example, in social networks, a social group may have several communities originated from different relations (e.g., friends, colleagues, etc.) or interests (e.g., sports, games, etc.) [12]. And a molecular graph may consist of various groups of atoms and bonds representing different functional units [13]. The complex relations among the multiple latent factors bring an urge for disentangling these factors

---

*Corresponding authors

35th Conference on Neural Information Processing Systems (NeurIPS 2021).

in contrastive graph representation learning, which remains unexplored by the existing holistic works. As a result, the graph representations learned by the existing methods contain a mixture of entangled factors, harming interpretability and leading to suboptimal performance for predictive tasks involving whole graph representations.

In this paper, we propose to learn disentangled contrastive graph representation, for the first time. Although disentangled representation learning, which aims to characterize the various underlying explanatory factors behind the observed data in different parts of the factorized representations [14, 15], has been demonstrated to be more explainable [12] and generalizable [14], disentangled graph contrastive learning faces the following two challenges. (1) Tailored graph encoder for disentangled contrastive learning. The graph encoder should be carefully designed so that it can be sufficiently expressive to infer the disentangled latent factors in the graph. (2) Tailored discrimination tasks designed for disentangled graph contrastive learning. Since task-dependent labels are not available in the self-supervised setting, disentangled graph contrastive learning can only utilize the limited amount of self-supervision information. This implies that the discrimination tasks should be well-designed for disentangled contrastive representation learning on graphs.

To tackle these challenges, we propose a novel disentangled graph contrastive learning model (**DGCL**) capable of disentangled contrastive learning on graphs. In particular, we first design a disentangled graph encoder whose key ingredient is a multi-channel message-passing layer. Each channel is tailored to aggregate features only from one disentangled latent factor. Then a separate readout operation in each channel summarizes the specific aspect of the graph according to the corresponding latent factor, so as to produce the disentangled graph representation. Next, we conduct contrastive learning in each representation subspace characterized by each factor independently instead of in the whole representation space. This novel factor-wise contrastive approach can ensure that each disentangled factor of the vectorized representations is sufficiently discriminative only under one specific aspect of the whole graph. Thus the representations are encouraged to be disentangled and best characterize the aspect pertinent to a latent factor of the graph. Compared with the existing methods, our proposed **DGCL** model encodes a graph with multiple disentangled representations, making it possible to explore the meaning of each channel, which benefits in more explainability for producing graph representations.

We conduct extensive experiments on both synthetic graph dataset and empirical well-known graph benchmarks. The results show that the representations learned from **DGCL** can achieve substantial performance gains on the downstream graph classification task compared with various state-of-the-art baselines.

The contributions of this paper are summarized as follows:

- We propose a disentangled graph contrastive learning model (**DGCL**), which is able to learn disentangled graph representation via factor-wise contrastive learning. To the best of our knowledge, we are the first to study disentangled self-supervised graph representation learning.

- We propose a disentangled graph encoder to capture multiple aspects of graphs through learning disentangled latent factors on graphs. We further present the factor-wise contrastive learning approach on tailored discrimination tasks in terms of each latent factor independently.

- We conduct extensive experiments to verify the efficacy of our proposed model for the graph classification task. The results on several graph classification datasets demonstrate that **DGCL** achieves state-of-the-art performance by significantly outperforming the baselines.

We introduce the problem formulation and preliminaries in Section 2. In Section 3, we describe the details of our proposed method. Section 4 presents the experimental results, including quantitative and qualitative comparisons. We review the related work in Section 5. Finally, we conclude our work in Section 6.

## 2 Problem Formulation and Preliminaries

### 2.1 Problem Formulation

Let $\mathbf{G} = \{G_i\}_{i=1}^N$ be a graph dataset with $N$ graphs. The key of most self-supervised graph representation learning methods, including ours, is to derive a graph encoder $f(\cdot)$, which outputs a

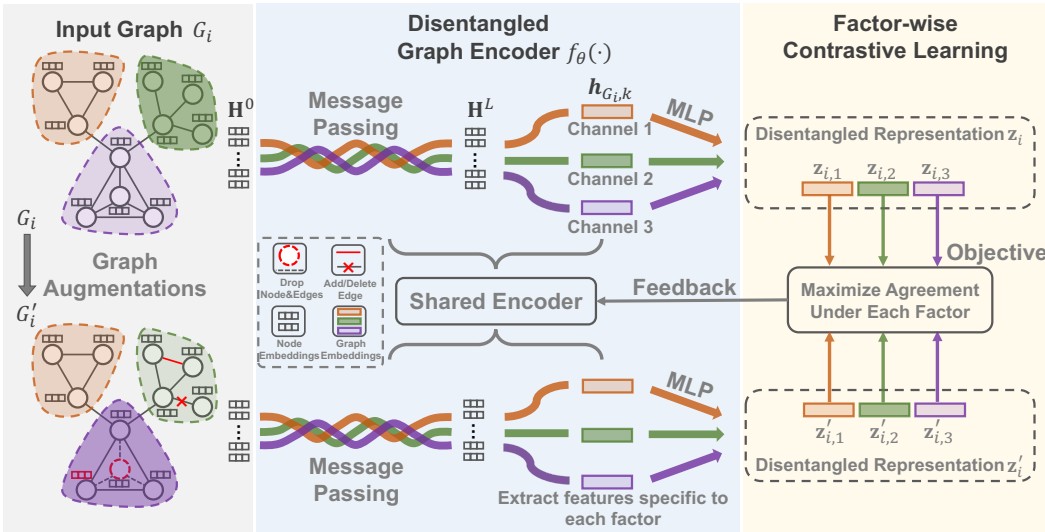

Figure 1: The framework of **DGCL** Model. (1) The input graph $G_i$ undergoes graph augmentations to produce $G'_i$, and both of them are fed into the shared disentangled graph encoder $f_\theta(\cdot)$. (2) In the encoder $f_\theta(\cdot)$, the node features $\mathbf{H}^0$ are first aggregated by $L$ message-passing layers and then taken as the input of a multi-channel message-passing layer. (3) Based on the disentangled graph representation $\mathbf{z}_i$, the factor-wise contrastive learning aims to maximize the agreement under each latent factor and provide feedback for the encoder to improve the disentanglement. This example assumes that there are three latent factors, hence the three channels.

$d$-dimensional representation $\mathbf{z}_i = f(G_i) \in \mathbb{R}^d$ for each input graph, such that $\mathbf{Z} = \{\mathbf{z}_i\}_{i=1}^N$ best describes $\mathbf{G}$. In this work, we aim to learn a multi-channel graph encoder $f_\theta(\cdot)$ with parameters $\theta$, so that the output $\mathbf{z}_i$ can be a disentangled representation, i.e. $f_\theta(\cdot) = \{f_\theta^{(k)}(\cdot)\}_{k=1}^K$, where $K$ is the number of channels. To be specific, $\mathbf{z}_i$ is expected to be composed of $K$ independent components, i.e., $\mathbf{z}_i = [\mathbf{z}_{i,1}, \mathbf{z}_{i,2}, \ldots, \mathbf{z}_{i,K}]$, where $\mathbf{z}_{i,k} = f_\theta^{(k)}(G_i) \in \mathbb{R}^{\Delta d}$, $k \in [1, K], \Delta d = d/K$, assuming that there are $K$ latent factors behind the graph instances to be disentangled. The $k^{\text{th}}$ component $\mathbf{z}_{i,k}$ is for characterizing the aspect of $G_i$ that is pertinent to factor $k$ accurately. We also assume that the value of $\mathbf{z}_{i,k}$ will be merely a white noise vector if the input graph $G_i$ does not contain any information of factor $k$. Note that we focus primarily on undirected graphs in our method, although it can be straightforwardly extended to directed graphs.

## 2.2 Preliminaries on Contrastive Learning

Unlike generative models, contrastive learning is an instance-wise discriminative approach that aims at making similar instances closer and dissimilar instances far from each other in representation space [16, 17]. It treats each instance in the dataset as a distinct class of its own and trains a classifier to distinguish between individual instance classes [18, 19]. Given a dataset $\mathbf{X} = \{x_i\}_{i=1}^N$, each instance $x_i$ is assigned with a unique surrogate label $y_i$, since no ground-truth labels are given. $y_i$ is often regarded as the ID of the instance in the dataset, i.e., $y_i = i$. So the probability classifier is defined as:

$$p_\theta(y_i|x_i) = \frac{\exp \phi(\mathrm{v}_i, \mathrm{v}'_{y_i})}{\sum_{j=1}^N \exp \phi(\mathrm{v}_i, \mathrm{v}'_{y_j})}, \tag{1}$$

where $\theta$ denotes the parameters of the encoder. Both $\mathrm{v}_i$ and $\mathrm{v}'_{y_i}$ are the embeddings from $x_i$, which are generated from two different encoders [20], or a shared encoder [21]. Before being passed into the encoder, the input $x_i$ could undergo data augmentations [21], which play a critical role in defining effective predictive tasks for learning the encoder. $\phi$ is the similarity function, often adopting cosine similarity with temperature $\tau$ [22], i.e., $\phi(\mathrm{v}_i, \mathrm{v}'_{y_i}) = \mathrm{v}'^\top_{y_i} \mathrm{v}_i/\tau$, assuming the embeddings are $\ell^2$-normalized. Then the learning objective is to maximize the joint probability $\prod_{i=1}^N p(y_i|x_i)$ over the dataset, namely minimize the negative log-likelihood function $\sum_{i=1}^N \ell_i$, if let $\ell_i = -\log p(y_i|x_i)$.

Note that loss $\ell_i$ could be NCE loss [18], InfoNCE loss [23], or NT-Xent loss [21]. The encoder will be encouraged to learn a representation space where samples (e.g., augmented data) from the same instance (e.g., an image, a graph) are pulled closer and samples from different instances are pushed apart [16]. For convenience, we follow the settings above in this work.

## 3 Disentangled Graph Contrastive Learning

In this section, we present the proposed **DGCL** model. The framework of **DGCL** is shown in Figure 1. In Section 3.1, we introduce the disentangled graph encoder to identify the complex latent factors and capture multiple aspects of graphs. Then in Section 3.2, we propose a factor-wise contrastive learning approach to conduct instance discrimination under each latent factor independently. The objective and the details of the optimization are derived in Section 3.3 and Section 3.4, respectively.

### 3.1 Disentangled Graph Encoder

The key of the disentangled graph encoder is to produce the factorized graph representation $\mathbf{z}_i = [\mathbf{z}_{i,1}, \mathbf{z}_{i,2}, \ldots, \mathbf{z}_{i,K}]$ for each input graph $G_i \in \mathbf{G}$. Based on the factorized representation, we can infer the related latent factors behind the graph.

Generally, GNNs use the graph structure and node features to learn the representation vector $\boldsymbol{h}_v$ of each node $v$ with a message-passing mechanism, i.e., iteratively updating the representation of a node by aggregating representations of its neighbors. The propagation of the $l^{\text{th}}$ layer is formulated as:

$$\boldsymbol{h}_v^l = \text{COMBINE}^l(\boldsymbol{h}_v^{l-1}, \text{AGGREGATE}^l(\{\boldsymbol{h}_u^{l-1} : u \in \mathcal{N}(v)\})), \tag{2}$$

where $\boldsymbol{h}_v^l$ is the representation of node $v$ at the $l^{\text{th}}$ layer and $\boldsymbol{h}_v^0$ is initialized with node features. $\mathcal{N}(v)$ is the neighborhood to node $v$. We use the term GNN to indicate the message-passing layer in Eq. (2).

Let $\mathbf{H}^l = \{\boldsymbol{h}_v^l | v \in V\}$ be the node embeddings after the $l^{\text{th}}$ GNN, where $V$ denotes the node set of the graph. After applying $L$ traditional message-passing layers, we propose a graph-disentanglement layer to learn the disentangled representations. The goal is to extract features specific to each latent factor with a separate channel. Specifically, we adopt $K$ separate channels to identify the complex heterogeneous latent factors and capture multiple aspects of the input graph. For each channel, we first utilize a $\text{GNN}_k$ to propagate information with its own parameters: $\mathbf{H}_k^{L+1} = \text{GNN}_k(\mathbf{H}^L, A)$, where $A$ is the adjacency matrix of the input graph. $\mathbf{H}_k^{L+1}$ is the node embeddings which is only pertinent to the $k^{\text{th}}$ latent factor. Then the READOUT function (i.e., pooling function) of each channel is used to summarize all the obtained node representations into a fixed-length graph-level representation: $\boldsymbol{h}_{G_i,k} = \text{READOUT}_k(\{\mathbf{H}_k^{L+1}\})$. Finally, each channel outputs the factorized graph representation with a separate MLP: $\mathbf{z}_{i,k} = \text{MLP}_k(\boldsymbol{h}_{G_i,k})$.

Compared with the existing graph encoders that are inherently holistic, our disentangled graph encoder consists of $K$ channels, rendering the possibility to identify the complex heterogeneous latent factors and capture multiple aspects of graphs.

### 3.2 Disentangled Factor-wise Contrastive Learning

Unlike the existing contrastive learning methods, **DGCL** designs a novel factor-wise instance discriminative task and learns to solve this task under each latent factor independently. This design not only makes similar samples closer and dissimilar samples far from each other in the representation space, but also encourages the learned representation to incorporate factor-level information for disentanglement.

Specifically, we assume that the formation of real-world graphs is usually driven by multiple latent heterogeneous factors. So the instance discriminative task should be represented as the expectation of several subtasks under the latent factors:

$$p_\theta(y_i | G_i) = \mathbb{E}_{p_\theta(k|G_i)} \left[ p_\theta(y_i | G_i, k) \right]. \tag{3}$$

Here $p_\theta(k|G_i)$ is the probability distribution over latent factors for the input graph $G_i$. $p_\theta(y_i | G_i, k)$ denotes the instance discrimination subtask under the $k^{\text{th}}$ latent factor.

Firstly, given the representation $\mathbf{z}_i$ of $G_i$ derived from the disentangled graph encoder $f_\theta(\cdot)$, we present a prototype-based method to obtain $p_\theta(k|G_i)$. We introduce $K$ latent factor prototypes $\{\mathbf{c}_k\}_{k=1}^K$, and the probability of the $k^{\text{th}}$ latent factor reflected in $G_i$ is parameterized as:

$$p_\theta(k|G_i) = \frac{\exp \phi(\mathbf{z}_{i,k}, \mathbf{c}_k)}{\sum_{k=1}^K \exp \phi(\mathbf{z}_{i,k}, \mathbf{c}_k)}, \tag{4}$$

where $\phi$ is the cosine similarity with temperature $\tau$, i.e., $\phi(\mathbf{a}, \mathbf{b}) = \text{COSINE}(\mathbf{a}, \mathbf{b})/\tau$ and $\text{COSINE}(\mathbf{a}, \mathbf{b}) = \mathbf{a}^\top \mathbf{b}/(\|\mathbf{a}\|_2 \|\mathbf{b}\|_2)$.

Then, we define the instance discrimination subtask under the $k^{\text{th}}$ latent factor as:

$$p_\theta(y_i|G_i, k) = \frac{\exp \phi(\mathbf{z}_{i,k}, \mathbf{z}'_{y_i,k})}{\sum_{j=1}^N \exp \phi(\mathbf{z}_{i,k}, \mathbf{z}'_{y_j,k})}, \tag{5}$$

where $\mathbf{z}_{i,k}$ and $\mathbf{z}'_{y_i,k}$ are the disentangled representations produced by the shared graph encoder, and $y_i$ is the unique surrogate label (see Section 2) of the graph $G_i$. In our method, we follow [18, 19] to implement $y_i$ as the ID of the graph in the dataset, i.e., $y_i = i$. For notation convenience, we do not distinguish $y_i$ and $i$ hereafter when there is no risk of confusion.

Next, we describe the process to get $\mathbf{z}'_{i,k}$, i.e. $\mathbf{z}'_{y_i,k}$ in Eq. (5). First, the input graph $G_i$ undergoes graph data augmentations to obtain its correlated views $G'_i$, and they form a positive pair. Data augmentation is expected to create novel and realistically rational data by applying certain transformations that do not affect the label, and plays a critical role in defining effective predictive tasks [21, 11]. We follow [11] to adopt four types of graph augmentation strategies, including node dropping, edge perturbation, attribute masking, and subgraph sampling. More details of graph augmentations can be found in supplementary material. Then, the augmented graph $G'_i$ is also fed into the shared disentangled graph encoder $f_\theta(\cdot)$ to produce $\mathbf{z}'_{i,k}$, i.e. $\mathbf{z}'_{y_i,k}$. Given the disentangled representations $\mathbf{z}_{i,k}$ and $\mathbf{z}'_{i,k}$ of the $G_i$ and $G'_i$ respectively, we conduct factor-wise contrastive learning for each latent factor independently as Eq. (5).

## 3.3 Evidence Lower Bound (ELBO)

We present the objective of our method. Following the existing methods [18, 19], we aim to maximize the joint probability $\prod_{i=1}^N p(y_i|G_i)$ over the graph dataset $\mathbf{G} = \{G_i\}_{i=1}^N$. We learn the model parameters $\theta$ by maximizing the log-likelihood:

$$\theta^* = \arg\max_\theta \sum_{i=1}^N \log p_\theta(y_i|G_i) = \arg\max_\theta \sum_{i=1}^N \log \mathbb{E}_{p_\theta(k|G_i)}[p_\theta(y_i|G_i, k)]. \tag{6}$$

However, directly maximizing the log-likelihood function is difficult because of the latent factors. Therefore, we instead optimize the evidence lower bound (ELBO) of the log-likelihood function given by Theorem 1. See the supplementary material for the proof.

**Theorem 1.** *The log likelihood function of each graph* $\log p_\theta(y_i|G_i)$ *is lower bounded by the ELBO:* $\mathcal{L}(\theta, i) = \mathbb{E}_{q_\theta(k|G_i, y_i)}[\log p_\theta(y_i|G_i, k)] - D_{KL}(q_\theta(k|G_i, y_i) \| p_\theta(k|G_i))$.

To make the ELBO as tight as possible, we require that $q_\theta(k|G_i, y_i)$ is close to $p_\theta(k|G_i, y_i)$, whose detailed implementations are provided in the next subsection (see Eq. (9) and Eq. (7)). In the ELBO $\mathcal{L}(\theta, i)$, $p_\theta(y_i|G_i, k)$ and $p_\theta(k|G_i)$ have been introduced in Eq. (5) and Eq. (4), respectively, and $q_\theta(k|G_i, y_i)$ is a variational distribution to infer the posterior distribution of the latent factors after observing both $G_i$ and its correlated view $G'_{y_i}$.

## 3.4 Optimization

We introduce a variational distribution $q_\theta(k|G_i, y_i)$ to infer the posterior probability $p_\theta(k|G_i, y_i)$ that is defined with Bayes' theorem as follows:

$$p_\theta(k|G_i, y_i) = \frac{p_\theta(k|G_i)p_\theta(y_i|G_i, k)}{\sum_{k=1}^K p_\theta(k|G_i)p_\theta(y_i|G_i, k)}. \tag{7}$$

$p_\theta(k|G_i, y_i)$ is the probability of the $k^{\text{th}}$ latent factor pertinent to both $G_i$ and the augmented $G_i'$ simultaneously. Compared with the prior distribution $p_\theta(k|G_i)$ in Eq. (4), $p_\theta(k|G_i, y_i)$ incorporates more useful information (i.e., factor-wise similarity) from $p_\theta(y_i|G_i, k)$. Although both $p_\theta(k|G_i)$ and $p_\theta(k|G_i, y_i)$ are designed to infer the latent factor distribution, $p_\theta(k|G_i)$ is calculated only given the graph $G_i$, but $p_\theta(k|G_i, y_i)$ is calculated after observing $G_i$, the augmented version $G_{y_i}'$, and their similarities under the specific latent factor.

However, we cannot compute the posterior probability tractably because of the term $p_\theta(y_i|G_i, k)$. If we directly calculate $p_\theta(y_i|G_i, k)$ according to Eq. (5), all the instances in the dataset $\mathbf{G}$ are needed for computing the denominator in Eq. (5) , which could be computationally prohibitive [24, 18, 19]. To tackle this obstacle, several strategies are proposed in the literature, including memory bank [18, 20], dynamic dictionary [25], NT-Xent loss [21]. Here, we adopt NT-Xent loss on a minibatch $\mathcal{B} \subseteq \mathbf{G}$. So in practice, the instance discrimination under each latent factor is calculated by:

$$\hat{p}_\theta(y_i|G_i, k) = \frac{\exp \phi(\mathbf{z}_{i,k}, \mathbf{z}_{i,k}')}{\sum_{j \in \mathcal{B}, j \neq i}^{|\mathcal{B}|} \exp \phi(\mathbf{z}_{i,k}, \mathbf{z}_{j,k}')}. \tag{8}$$

We approximate the posterior probability $p_\theta(k|G_i, y_i)$ with a variational distribution defined as:

$$q_\theta(k|G_i, y_i) = \frac{p_\theta(k|G_i)\hat{p}_\theta(y_i|G_i, k)}{\sum_{k=1}^{K} p_\theta(k|G_i)\hat{p}_\theta(y_i|G_i, k)}. \tag{9}$$

Finally, we seek to learn the parameters $\theta$ of the disentangled graph encoder. More specifically, we calculate $q_\theta(k|G_i, y_i)$ and maximize the ELBO over a mini-batch $\mathcal{B}$ using mini-batch gradient ascent:

$$\mathcal{L}(\theta, \mathcal{B}) = \sum_{i \in \mathcal{B}} \mathcal{L}(\theta, i). \tag{10}$$

**Encourage disentanglement.** Note that our objective and its optimization can inherently encourage disentanglement without adding extra regularization term (e.g., minimizing mutual information). The reason is that factorizing the instance discrimination into $K$ factor-wise subtasks will enforce the independence of the learned graph representation $\mathbf{z}_i$. Besides, $q_\theta(k|G_i, y_i)$ is computed based on $k^{\text{th}}$ and other $K-1$ latent factors. Thus, the graph encoder is forced to preserve exclusive information in each channel to get more accurate approximation to the posterior, if a tighter ELBO is expected. The strong inductive biases in **DGCL** encourage to learn disentangled graph representations that match the ground truth factors behind the graphs.

## 4 Experiments

We empirically evaluate our proposed method through experiments, and analyze its behavior on synthetic graph dataset to gain deeper insight. Ablation studies are conducted to show the effectiveness of the proposed method. We provide more discussions in supplementary material, including the complexity of our method, the impact of the hyper-parameters, etc.

### 4.1 Experimental Setup

**Datasets.** To demonstrate the advantages of our method, we conduct experiments on nine well-known graph classification datasets including four bioinformatics datasets, i.e., MUTAG, PTC-MR, NCI1, PROTEINS, and five social network datasets, i.e., COLLAB, IMDB-BINARY, IMDB-MULTI, REDDIT-BINARY, and REDDIT-MULTI-5K. We also adopt a larger graph dataset ogbg-molhiv from Open Graph BenchMark (OGB) [26]. More details are provided in supplementary material.

**Baselines.** We compare **DGCL** with the following two groups of baselines. One group of baselines are graph kernels including Shortest Path Kernel (SP) [27], Graphlet Kernel (GK) [28], Weisfeiler-Lehman Sub-tree Kernel (WL) [29], Deep Graph Kernels (DGK) [30], and Multi-Scale Laplacian Kernel (MLG) [31]. The other group of baselines are classical unsupervised graph representation learning methods including node2vec [32], sub2vec [33], graph2vec [34], GVAE [35], and more recent contrastive graph representation learning methods including InfoGraph [5], GCC [10], MVGRL [9], and GraphCL [11].

**Evaluation.** To verify the effectiveness of our method, we follow the common evaluation protocols in the existing literature [34, 5, 30, 11], where graph embeddings are generated from each method

Table 1: Graph classification accuracy (%) of **DGCL** and baselines. In each column, the boldfaced score denotes the best result and the underlined score represents the second-best result. "–" indicates the result is not reported in the paper.

| | MUTAG | PTC-MR | PROTEINS | NCI1 | IMDB-B | IMDB-M | RDT-B | RDT-M5K | COLLAB |
|---|---|---|---|---|---|---|---|---|---|
| SP | 85.2±2.4 | 58.2±2.4 | 75.1±0.5 | 73.0±0.2 | 55.6±0.2 | 38.0±0.3 | 64.1±0.1 | 39.6±0.2 | – |
| GK | 81.7±2.1 | 57.3±1.4 | 71.7±0.6 | 62.3±0.3 | 65.9±1.0 | 43.9±0.4 | 77.3±0.2 | 41.0±0.2 | 72.8±0.3 |
| WL | 80.7±3.0 | 58.0±0.5 | 72.9±0.6 | 80.0±0.5 | 72.3±3.4 | 47.0±0.5 | 68.8±0.4 | 46.1±0.2 | – |
| DGK | 87.4±2.7 | 60.1±2.6 | 73.3±0.8 | 80.3±0.5 | 67.0±0.6 | 44.6±0.5 | 78.0±0.4 | 41.3±0.2 | 73.1±0.3 |
| MLG | 87.9±1.6 | 63.3±1.5 | 76.1±2.0 | 80.8±1.3 | 66.6±0.3 | 41.2±0.0 | – | – | – |
| node2vec | 72.6±10.2 | 58.6±8.0 | 57.5±3.6 | 54.9±1.6 | – | – | – | – | – |
| sub2vec | 61.1±15.8 | 60.0±6.4 | 53.0±5.6 | 52.8±1.5 | 55.3±1.5 | 36.7±0.8 | 71.5±0.4 | 36.7±0.4 | – |
| graph2vec | 83.2±9.3 | 60.2±6.9 | 73.3±2.1 | 73.2±1.8 | 71.1±0.5 | 50.4±0.9 | 75.8±1.0 | 47.9±0.3 | – |
| GVAE | 87.7±0.7 | 61.2±1.8 | – | – | 70.7±0.7 | 49.3±0.4 | 87.1±0.1 | 52.8±0.2 | – |
| InfoGraph | 89.0±1.1 | 61.7±1.4 | 74.4±0.3 | 76.2±1.1 | 73.0±0.9 | 49.7±0.5 | 82.5±1.4 | 53.5±1.0 | 70.7±1.1 |
| GCC | – | – | – | – | 72.0 | 49.4 | 89.8 | 53.7 | 78.9 |
| MVGRL | 89.7±1.1 | 62.5±1.7 | – | – | 74.2±0.7 | 51.2±0.5 | 84.5±0.6 | – | – |
| GraphCL | 86.8±1.3 | 63.6±1.8 | 74.4±0.5 | 77.9±0.4 | 71.1±0.4 | 50.7±0.4 | 89.5±0.8 | 56.0±0.3 | 71.4±1.2 |
| **DGCL** | **92.1±0.8** | **65.8±1.5** | **76.4±0.5** | **81.9±0.2** | **75.9±0.7** | **51.9±0.4** | **91.8±0.2** | **56.1±0.2** | **81.2±0.3** |

and then fed into a downstream SVM classifier. We adopt the 10-fold cross validation accuracy, and report the mean accuracy (%) with standard variation after five repeated runs.

**Implementation Details.** For a fair comparison, the hyper-parameters of the graph augmentations are kept consistent with GraphCL. We use GIN [4] as the message-passing layers since it is shown to be one of the most expressive message-passing GNNs. Since the ground-truth number of the latent factors is unknown, we search the number of channels $K$ from 1 to 10. More implementation details can be found in supplementary material.

## 4.2 Results on Real Benchmark Graphs

The results are reported in Table 1. We can see that the graph contrastive learning methods generally outperform the graph kernel methods or the classical unsupervised methods, which verify the effectiveness of contrastive learning. Our method **DGCL** consistently achieves the best performance compared with other contrastive methods (e.g., MVGRL, GrpahCL) and classical unsupervised methods (e.g., graph2vec, GVAE), demonstrating the superiority of our method. For example, our method increases the classification accuracy by 2.4%, 2.2%, and 2.0% compared with the second-best methods on MUTAG, PTC-MR, and RDT-B, respectively. We attribute the results to the fact that these existing methods fail to identify the underlying latent factors which are important in preserving graph properties and can not learn the disentangled representations. In contrast, we disentangle graph representations to explicitly consider the entanglement of heterogeneous factors. When compared to graph kernel methods, our method also has the best accuracy on all the datasets. Notice that none of these kernel methods is consistently competitive across all of the datasets, as opposed to our method.

Besides the common setting of unsupervised representation learning above, we also consider another setting of semi-supervised representation learning [11] on the ogbg-molhiv from Open Graph Benchmark [26] to better evaluate our method. Specifically, we first perform pre-training with all training data without labels. Then we conduct fine-tuning on the partial labeled training data and evaluation on the validation/test sets. The task on the ogbg-molhiv dataset is binary classification evaluated by ROC-AUC metric, instead of accuracy in the experiments above. We adopt the provided evaluator and dataset splits for a fair comparison. We compare **DGCL** with the strong self-supervised baseline GraphCL with 1%, 10%, and 20% label rate for fine-tuning. The results are shown in Figure 2. With the increase of the label rate, the results improve for both our method and GraphCL. Our method achieves significant improvement over GraphCL with 1.4%, 3.0%, and 1.4% performance gains at 1%, 10%, and 20% label rate, respectively. The results illustrate that our method is also able to handle large-scale graphs, demonstrating the benefit of learning disentangled graph representations in the contrastive manner.

## 4.3 Results on Synthetic Graphs

To further investigate the behavior of our method, we generate a synthetic dataset consisting of 1,000 graphs with known latent factors. Specifically, we generate synthetic graphs using the stochastic

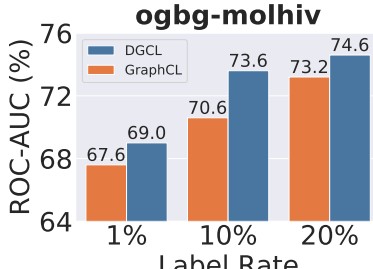
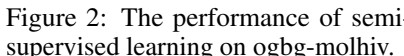
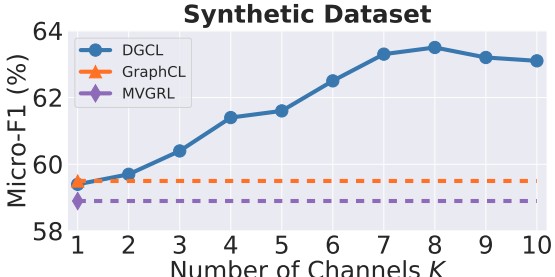

Figure 2: The performance of semi-supervised learning on ogbg-molhiv.

Figure 3: Micro-F1 (%) of two baselines and our **DGCL** with different number of channels $K$.

block model [36]. Each graph contains four communities and each community consists of 10 nodes. We define the latent factor as the probability $p$ that two nodes are connected in a community. $p$ can take value from $\{0.2, 0.3, \dots, 0.9\}$, meaning that there are eight latent factors in the dataset in total. The probability for each community is drawn from the eight possible choices without replacement. Two nodes in different communities are connected with probability $0.05$. The rows of the adjacency matrices are used as node features, and the ground-truth communities are used as labels, i.e., there are 8 classes and each graph has 4 labels. We train our method and two baseline methods, i.e., MVGRL and GraphCL on the generated synthetic dataset with self-supervision. Then we adopt the SVM classifier on the learned graph embeddings for each method and use the Micro-F1 (%) as the evaluation metric. Other settings are the same as Section 4.1.

We vary the number of channels $K$ of our method and report the results in Figure 3. Our method reports better performance than the baselines. We also find that as $K$ increases from 1 to 8, the result of **DGCL** improves, which verifies the importance of disentangling latent factors. After reaching the peak at $K = 8$, the performance slightly drops, but in general, our method is not very sensitive when $K$ is not too large. Our method achieves the best results when $K$ is equal to the ground-truth number of latent factors, indicating we can model the underlying structure of this simulation dataset.

Besides the quantitative evaluation, we also provide a qualitative evaluation by plotting the correlation of the latent features in Figure 4. The figure shows the absolute values of the correlation between the elements of 128-dimensional graph representation obtained from MVGRL, GraphCL, and our **DGCL** ($K = 8$) on the synthetic dataset. We can see from the results that the graph representations of MVGRL and GraphCL are entangled. In comparison, the correlation of our **DGCL** shows eight diagonal blocks, meaning that the channels of **DGCL** likely extract mutually exclusive information and output disentangled representations.

## 4.4 Ablation Studies

We perform ablation studies over the key components of our method to understand their functionalities more deeply. We compare **DGCL** with the following two variants: (1) Variant 1: it sets $p_\theta(k|G_i) = 1/K$ a uniform distribution of latent factors. (2) Variant 2: it sets $K = 1$ directly, so that our method will degenerate to the entangled graph contrastive learning model.

Table 2: Ablation studies on the variants of our method. We report the accuracy (%) with standard variation on the datasets. The results of Variant 1 and 2 drop compared with **DGCL**, demonstrating the significance to infer latent factors behind the graphs and conduct factor-wise contrastive learning.

|  | MUTAG | PTC-MR | PROTEINS | NCI1 | IMDB-B | IMDB-M | RDT-B | RDT-M5K | COLLAB |
|---|---|---|---|---|---|---|---|---|---|
| DGCL | **92.1±0.8** | **65.8±1.5** | **76.4±0.5** | **81.9±0.2** | **75.9±0.7** | **51.9±0.4** | **91.8±0.2** | **56.1±0.2** | **81.2±0.3** |
| Variant 1 | 89.3±0.3 | 64.3±1.3 | 74.9±0.2 | 78.5±0.5 | 73.4±0.5 | 50.3±0.2 | 91.1±0.7 | 55.9±0.3 | 77.5±0.4 |
| Variant 2 | 86.5±0.6 | 63.5±1.6 | 73.9±0.6 | 77.7±0.6 | 70.9±0.5 | 49.8±0.3 | 89.7±0.6 | 55.7±0.2 | 71.5±0.8 |

The results of **DGCL** and its variants are shown in Table 2. We observe a drop in performance of Variant 1, demonstrating the efficacy of inferring the latent factors of the graphs. In variant 2, the latent factors are entangled in the graph representation, making difficulties for characterizing different aspects of the graphs and conducting discrimination tasks in terms of each latent factor independently.

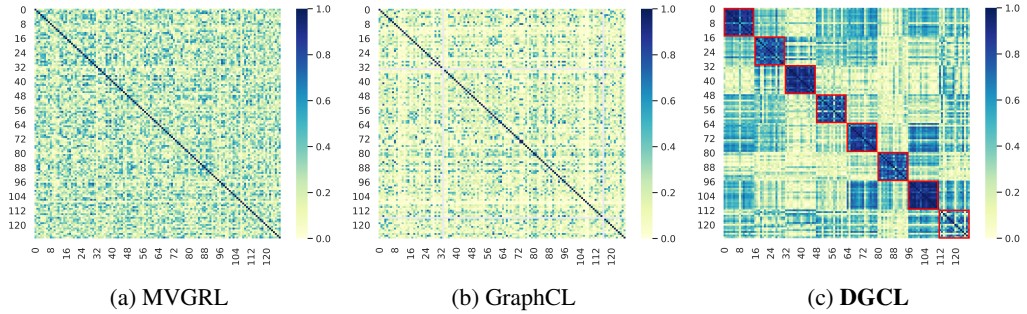

|            |            |            |
|:----------:|:----------:|:----------:|
| (a) MVGRL  | (b) GraphCL| (c) **DGCL** |

Figure 4: An analysis of feature correlation on the synthetic graphs with eight latent factors. The figures show the absolute value of the correlations between the elements of the representations learned by MVGRL, GraphCL, and **DGCL** with eight channels, respectively. We can see that the representations generated from **DGCL** present a more block-wise correlation pattern, indicating that the eight channels of the disentangled graph encoder in **DGCL** are able to capture mutually exclusive information and the latent features have indeed been disentangled.

The deterioration of performance verifies the significance of the proposed factor-wise contrastive learning.

## 5   Related Work

In this section, we review related works on graph neural networks, contrastive learning on graphs, and disentangled representation learning.

**Graph Neural Networks.**   Graph structured data is ubiquitous in the real world [26, 37, 38, 39, 40]. Recently, graph neural networks (GNNs) [1, 2, 3, 4, 41, 42] have revolutionized the field of graph representation learning. GNNs generally adopt a neighborhood aggregation (message passing) paradigm, i.e., the embedding of node is iteratively updated by aggregating embeddings of its neighbors [3, 4]. The representation of the whole graph is summarized on node embeddings through the readout function, i.e., graph pooling [4]. However, for achieving state-of-the-art performance, most famous GNNs, including their variants, are trained end-to-end with task-specific labels, which could be extremely scarce for some graph datasets. Compared with these supervised models, our model, based on self-supervised contrastive learning, can largely eliminate the over-dependence on the manual labels, which is crucial for graph representation learning.

**Contrastive Learning on Graphs.**   Recently, contrastive learning, adopting the instance discrimination as the pretext task, has become a dominant component in self-supervised learning methods [25, 21, 18, 19, 23]. Some literatures utilizing contrastive learning for graph data are proposed [5, 10, 9, 11, 43, 7]. The key of these methods is to maximize the agreement (i.e., similarity) between proper transformations or different views of the input graph. However, the existing graph contrastive learning methods explore general settings where entanglement is severe and do not incorporate disentangled representation learning. They fail to recognize and disentangle the heterogeneous latent factors behind complex graph data. These holistic methods have limited capacity in preserving detailed graph properties, which easily result in suboptimal representations for downstream tasks.

**Disentangled Representation Learning.**   Disentangled representation learning is to learn factorized representations that identify and disentangle the underlying explanatory factors hidden in the observed data [14]. The existing efforts about disentangled representation learning are mainly on computer vision [44, 45, 46]. It has been raising a surge of interest in graph-structured data recently [47, 48, 49, 12]. This line of works attempts to learn disentangled representations for graphs but heavily relies on the annotated labels, which largely restricts their applications where labeled data are unavailable or expensive to collect. On the other hand, some works [35, 50] are based on the generative model, namely utilizing Variational Autoencoders (VAEs) on graph for disentanglement, since the hyper-parameter $\beta$ of VAEs can balance the reconstruction and disentanglement [51, 52]. However, the reconstruction in generative methods could be computationally expensive [53, 17] and even introduce bias that has a negative effect on the learned representation [23]. In addition, the reconstruction for

graph-structured data often involves discrete decisions that are not differentiable [50]. How to learn disentangled representation on graph-structured data with contrastive learning is largely unexplored.

## 6 Conclusions

In this paper, we propose a disentangled graph contrastive learning model (**DGCL**) to solve the problem of disentangled self-supervised graph representation learning for the first time. We design a disentangled graph encoder with a tailored multi-channel message-passing layer, which is capable of aggregating features in a disentangled manner. We further propose a factor-wise contrastive learning approach to solve instance discrimination task under each latent factor independently, so that the learned representations of **DGCL** are encouraged to not only best describe the graphs but also be disentangled. Therefore, each component of the disentangled representations tends to characterize a disentangled aspect of the graph that is pertinent to a latent factor. Extensive experiments on both synthetic and real-world datasets demonstrate the superiority of our method against several state-of-the-art baselines in self-supervised graph representation learning.

## Acknowledgments

This work is supported by the National Key Research and Development Program of China No. 2020AAA0106300 and National Natural Science Foundation of China (No. 62050110, No. 62102222).

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
