# Disentangled Contrastive Learning on Graphs
## (Supplementary Material)

**Haoyang Li[1], Xin Wang[1], Ziwei Zhang[1], Zehuan Yuan[2], Hang Li[2], Wenwu Zhu[1]**
[1]Tsinghua University, [2]Bytedance
lihy18@mails.tsinghua.edu.cn, {xin_wang, zwzhang}@tsinghua.edu.cn,
{yuanzehuan, lihang.lh}@bytedance.com, wwzhu@tsinghua.edu.cn

## A  Training Details

### A.1  Hardware and Software Configurations

We conduct the experiments with:

- Operating System: Ubuntu 18.04.1 LTS
- CPU: Intel(R) Xeon(R) CPU E5-2699 v4@2.20GHz
- GPU: NVIDIA GeForce GTX TITAN X
- Software: Python 3.6.5; NumPy 1.18.0; PyTorch 1.7.0; PyTorch Geometric 1.6.1 [1].

### A.2  Datasets

The statistics of the datasets are in Table 1.

We adopt four bioinformatics datasets in the experiment. MUTAG dataset contains mutagenic aromatic and heteroaromatic nitro compounds. PTC dataset contains chemical compounds reported for carcinogenicity of rats. PROTEINS is a dataset where nodes are secondary structure elements, and there is a connection between two nodes if they are neighbors in the amino-acid sequence or in 3D space. NCI1 is a subset of balanced datasets of chemical compounds released by the National Cancer Institute (NCI).

We also conduct the experiment on five social network datasets. IMDB contains movies information, in which the nodes represent actors/actresses and the two nodes have connections if they have acted in the same movie. IMDB-BINARY consists of two genres of movies, while IMDB-MULTI contains movies from Comedy, Romance and Sci-Fi genres. Reddit datasets were created using threads in different subreddits, where nodes are users who responded to that particular thread and edges represent that one user responds to another user's comment. REDDIT-BINARY labels each graph as 2 labels and REDDIT-MULTI-5K labels graphs into 5 labels. COLLAB dataset is derived from 3 public collaboration datasets, i.e., High Energy Physics, Condensed Matter Physics and Astro Physics. Each graph corresponds to an ego-network of different researchers from each field. The labels denote the fields the corresponding researchers belong to.

Note that the node features are not provided for the five social network datasets. Therefore, we follow previous works [2, 3] to use a constant vector as the node features for REDDIT-BINARY and a one-hot encoding of node degrees as node features for the other datasets.

In addition, we use a larger graph dataset ogbg-molhiv from Open Graph Benchmark (OGB). Each graph in ogbg-molhiv represents a molecule, where nodes are atoms, and edges are chemical bonds. The input features of nodes are 9-dimensional, containing atomic number and chirality, as well as other additional atom features such as formal charge and whether the atom is in the ring or not.

The datasets are publicly available.

35th Conference on Neural Information Processing Systems (NeurIPS 2021).

Table 1: The statistics of the datasets.

|  | MUTAG | PTC-MR | PROTEINS | NCI1 | IMDB-B | IMDB-M | RDT-B | RDT-M5K | COLLAB | OGBG-MOLHIV |
|---|---|---|---|---|---|---|---|---|---|---|
| # graphs | 188 | 344 | 1,113 | 4,110 | 1,000 | 1,500 | 2,000 | 4,999 | 5,000 | 41,127 |
| # classes | 2 | 2 | 2 | 2 | 2 | 3 | 2 | 5 | 3 | 2 |
| Avg # nodes | 17.9 | 14.3 | 39.1 | 29.9 | 19.8 | 13.0 | 429.6 | 508.5 | 74.5 | 25.5 |
| Avg # edges | 19.8 | 14.7 | 72.8 | 32.3 | 96.5 | 65.9 | 497.8 | 594.9 | 2457.8 | 27.5 |

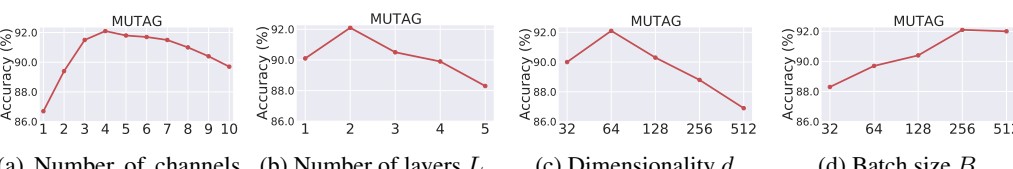

(a) Number of channels $K$.   (b) Number of layers $L$.   (c) Dimensionality $d$.   (d) Batch size $B$.

Figure 1: Impact of different hyper-parameters on MUTAG dataset.

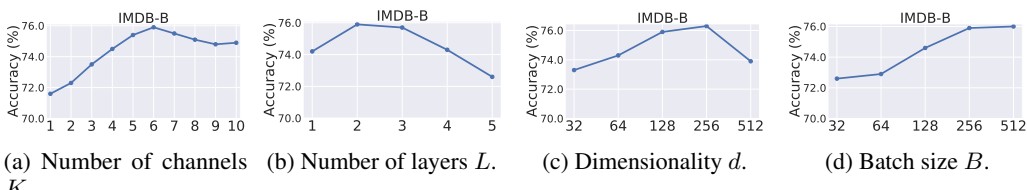

(a) Number of channels $K$.   (b) Number of layers $L$.   (c) Dimensionality $d$.   (d) Batch size $B$.

Figure 2: Impact of different hyper-parameters on IMDB-B dataset.

- ogbg-molhiv: `https://ogb.stanford.edu/docs/graphprop/` with MIT License
- The other nine graph datasets: `https://chrsmrrs.github.io/datasets/docs/datasets/` with license unspecified

## A.3  Graph Augmentation

The data augmentation is critical to contrastive learning methods. For a fair comparison, we follow the previous work GraphCL [4] and randomly perform one type of data augmentations for graphs as follows:

- **Node dropping.** Given the input graph, it will randomly discard 20% nodes along with their edges, implying that the missing nodes do not affect the model predictions much.

- **Edge perturbation.** Given the input graph, it will randomly add or cut a certain portion of connections between nodes with the probability of 0.2. This augmentation can prompt robustness of the graph encoder to the edge connectivity pattern variances.

- **Attribute masking.** It will set the feature of 20% nodes in the graph to Gaussian noises with mean and standard deviation is 0.5. The underlying prior is that missing part of the features do not affect the semantic information of the whole graph.

- **Subgraph sampling.** It will sample a subgraph, including 20% nodes from the input graph, using random walk. The assumption is that the semantic information of the whole graph can be reflected by its partial structure.

## A.4  Implementation Details

We implement our model in PyTorch. We adopt the Adam [5] optimizer, which is a variant of Stochastic Gradient Descent (SGD) with adaptive moment estimation. For a fair comparison, we follow the default setting in [4]. We use GIN [2] as the message-passing layers since it is shown to be one of the most expressive message-passing GNNs. Note that the ground-truth number of the latent factors is unknown, so we search the number of channels $K$ from 1 to 10. For the unsupervised setting, we use SVM as the downstream classifier. We adopt the 10-fold cross validation accuracy, and report the mean accuracy (%) with standard variation after five repeated runs. For the semi-supervised setting, we perform experiments with 1% , 10%, and 20% label rate.

**Algorithm 1** The training procedure of our method (**DGCL**).

---

**Input:** A graph dataset $\mathbf{G} = \{G_i\}_{i=1}^N$
**Output:** The disentangled representations $\mathbf{Z} = \{\mathbf{z}_i\}_{i=1}^N$ for $\mathbf{G}$

1: **function** DISENTANGLEDENCODER($G_i$)
2:     **for** $l \leftarrow 1$ to $L$ **do**
3:         $\mathbf{H}^l = \text{GNN}^l(\mathbf{H}^{l-1}, A)$
4:     **end for**
5:     **for** $k \leftarrow 1$ to $K$ **do**                                     ▷ separate $K$ channels
6:         $\mathbf{H}_k^{L+1} = \text{GNN}_k(\mathbf{H}^L, A)$
7:         $\mathbf{z}_{i,k} = \text{MLP}_k(\text{READOUT}_k(\{\mathbf{H}_k^{L+1}\}))$
8:     **end for**
9:     **return** $\mathbf{z}_i$                                     ▷ disentangled graph representation
10: **end function**
11: **for** sampled minibatch $\mathcal{B} = \{G_i\}_{i=1}^{|\mathcal{B}|}$ **do**
12:     **for** $G_i \in \mathcal{B}$ **do**                                ▷ disentangled graph encoding
13:         $\mathbf{z}_i = $ DISENTANGLEDENCODER($G_i$)
14:         $G_i' = $ GRAPHAUGMENTATION($G_i$)                         ▷ augmentation
15:         $\mathbf{z}_i' = $ DISENTANGLEDENCODER($G_i'$)
16:         Calculate $p_\theta(k|G_i)$ by Eq. (4)
17:     **end for**
18:     **for** $k \leftarrow 1$ to $K$ **do**                           ▷ factor-wise contrastive learning
19:         **for** $i \leftarrow 1$ to $|\mathcal{B}|$ and $j \leftarrow 1$ to $|\mathcal{B}|$ **do**
20:             $s_{i,j}^{(k)} = \phi(\mathbf{z}_{i,k}, \mathbf{z}_{j,k}')$                       ▷ similarity under $k^{\text{th}}$ factor
21:         **end for**
22:         Calculate $\hat{p}_\theta(y_i|G_i, k) = \exp s_{i,i}^{(k)} / \sum_{j=1, j\neq i}^{|\mathcal{B}|} \exp s_{i,j}^{(k)}$
23:     **end for**
24:     **for** $G_i \in \mathcal{B}$ **do**                                ▷ optimization objective
25:         Calculate $q_\theta(k|G_i, y_i)$ by Eq. (9)
26:         Calculate ELBO $\mathcal{L}(\theta, i)$
27:     **end for**
28:     Calculate ELBO over a minibatch $\mathcal{L}(\theta, \mathcal{B})$
29:     Update $\theta$ to maximize $\mathcal{L}(\theta, \mathcal{B})$, using the gradient $\nabla_\theta \mathcal{L}(\theta, \mathcal{B})$
30: **end for**
31: $\mathbf{Z} = \{\mathbf{z}_i\}_{i=1}^N$, where $\mathbf{z}_i = $ DISENTANGLEDENCODER($G_i$), $G_i \in \mathbf{G}$

---

We list the detailed training procedure of our method in Algorithm 1.

# B   Hyper-parameter Sensitivity

We investigate the sensitivity of hyper-parameters of our method: the number of channels $K$, the number of message-passing layers $L$, the dimensionality of the embeddings $d$, and the batch size $B$. Among them, the number of channels $K$ is the most important hyper-parameter. For simplicity, we only report the results on the MUTAG (Figure 1) and IMDB-B (Figure 2) datasets, while the results on other datasets show similar patterns. From Figures 1 and 2, we can observe that the performance increases at first with a larger $K$ and drops after reaching a peak, showing that a proper number of channels $K$ matching the real latent factors behind the observed data can lead to better results. Then, the number of message-passing layers $L$ is also important because the graph model with a small $L$ has the limited model capacity and may not be able to fuse enough information from neighbors, and a very large $L$ could also lead to the over-smoothing problem [6]. In addition, the optimal dimensionality of embeddings $d$ for MUTAG is relatively smaller than that for IMDB-B, since the former dataset only consists of 188 graphs but the latter contains 1,000 graphs with more nodes and edges. A too large $d$ may induce over-fitting and hurt the performance. Finally, we find that our method benefits from larger batch sizes, which is consistent with the common phenomenon in contrastive learning [7].

## C Evidence Lower Bound (ELBO)

**Theorem 1.** *The log likelihood function of each graph* $\log p_\theta(y_i|G_i)$ *is lower bounded by the ELBO:*
$\mathcal{L}(\theta, i) = \mathbb{E}_{q_\theta(k|G_i, y_i)}[\log p_\theta(y_i|G_i, k)] - D_{KL}(q_\theta(k|G_i, y_i) \parallel p_\theta(k|G_i)).$

*Proof.*

$$
\begin{aligned}
&\log p_\theta(y_i|G_i) \\
&= \mathbb{E}_{q_\theta(k|G_i,y_i)}\left[\log p_\theta(y_i|G_i)\right] \\
&= \mathbb{E}_{q_\theta(k|G_i,y_i)}\left[\log \frac{p_\theta(y_i,k|G_i)}{p_\theta(k|G_i,y_i)}\right] \\
&= \mathbb{E}_{q_\theta(k|G_i,y_i)}\left[\log \frac{p_\theta(y_i,k|G_i)}{q_\theta(k|G_i,y_i)}\frac{q_\theta(k|G_i,y_i)}{p_\theta(k|G_i,y_i)}\right] \\
&= \mathbb{E}_{q_\theta(k|G_i,y_i)}\left[\log \frac{p_\theta(y_i,k|G_i)}{q_\theta(k|G_i,y_i)}\right] + \mathbb{E}_{q_\theta(k|G_i,y_i)}\left[\log \frac{q_\theta(k|G_i,y_i)}{p_\theta(k|G_i,y_i)}\right] \\
&= \mathbb{E}_{q_\theta(k|G_i,y_i)}\left[\log \frac{p_\theta(y_i,k|G_i)}{q_\theta(k|G_i,y_i)}\right] + D_{KL}(q_\theta(k|G_i,y_i) \parallel p_\theta(k|G_i,y_i)) \\
&\geq \mathbb{E}_{q_\theta(k|G_i,y_i)}\left[\log \frac{p_\theta(y_i,k|G_i)}{q_\theta(k|G_i,y_i)}\right] \\
&= \mathbb{E}_{q_\theta(k|G_i,y_i)}\left[\log p_\theta(y_i|G_i,k)\frac{p_\theta(k|G_i)}{q_\theta(k|G_i,y_i)}\right] \\
&= \mathbb{E}_{q_\theta(k|G_i,y_i)}\left[\log p_\theta(y_i|G_i,k)\right] - D_{KL}(q_\theta(k|G_i,y_i) \parallel p_\theta(k|G_i)) \\
&= \mathcal{L}(\theta, i).
\end{aligned}
\tag{1}
$$

The equality holds when $D_{KL}(q_\theta(k|G_i,y_i) \parallel p_\theta(k|G_i,y_i)) = 0$. Note that in the third-to-last line above, we have used $p_\theta(y_i, k|G_i) = p_\theta(k|G_i)p_\theta(y_i|G_i, k)$. $\qquad\square$

## D Complexity Analysis

The time complexity of our method is $O(M)$, where $M$ denotes the number of edges in the graphs. Specifically, **DGCL** adopts GIN as the message-passing layers so the time complexity of the disentangled graph encoder is $O(M)$. As for the factor-wise contrastive learning, the positive and negative samples are drawn from graph data augmentations and graphs from the same minibatch respectively, which will not induce a higher computational cost. The time complexity of self-supervised learning baselines (e.g., GraphCL, MVGRL, etc.) in the experiments is also $O(M)$. Therefore, the time complexity of our proposed **DGCL** is on par with these baselines. Notice that GVAE, one of the unsupervised baselines, has $O(N^2)$ time complexity due to the adjacency matrix reconstruction in the VAE framework, where $N$ denotes the number of nodes in the graphs. Since $O(M) \ll O(N^2)$ for sparse graphs, our proposed method is much more scalable than GVAE.

## E Number of Parameters

For our method, the number of parameters is $O(Ld^2 + K * (d/K))$, i.e., $O(Ld^2)$, where $K$ is the number of channels, $L$ is the number of message-passing layers of disentangled graph encoder, and $d$ is the dimensionality of the embeddings. Specifically, because we adopt GIN as the message passing layers, the number of parameters of the disentangled graph encoder is $O(Ld^2)$. The number of parameters of $K$ latent factor prototypes is $(K * (d/K))$. For the baselines using Graph Neural Network (i.e., GCN or GIN) as the graph encoder, including GVAE, InfoGraph, GCC, MVGRL, and GraphCL, the number of parameters is $O(Ld^2)$. Therefore, the number of parameters of the proposed method and the baselines are comparable.

## F   Future Direction

In this work, we focus on the disentangled graph contrastive learning for graph-level representations. So the proposed method cannot be directly applied to the node-level representation tasks. Nevertheless, we think a similar methodology of our method could be extended to node-level tasks in the future.