# OpenReview forum: "Disentangled Contrastive Learning on Graphs"
_NeurIPS.cc/2021/Conference — NeurIPS 2021 Poster_

### Official Review · Reviewer_ubo5 · 2021-07-03

**Rating:** 6
**Confidence:** 4

**Summary:**

This paper propose a disentangled graph contrastive learning model to solve the graph representation learning task. The proposed model uses a disentangled graph encoder with a tailored multi-channel message passing layer, where the EM algorithm is used to optimize the parameters of the model. The authors conduct experiment with their models with both synthetic and real-world datasets, and the result validate the effectiveness of the proposed model.


**Ethics Review Area:**

["I don’t know"]

**Limitations And Societal Impact:**

The authors don't describe the limitations of their work.
Suggestions to improve:
- It will be good if the authors can make their codes publicly available, as they promised.
- It will be good if the authors can do some case studies to show the disentanglement on the learned latent factors.

**Main Review:**

​1. This paper is well-written and easy to follow.

2. The model proposed model is basically an extension from the GraphCL [49] model, where latent factors for the input graph are introduced for learning the disentangled factor-level information, which is novel but limited. The approach of maximizing ELBO is generally sound.

3. Although the authors provide an analysis of feature correlation with the disentangled latent factors, the proposed approach for learning such latent factors is not well motivated. Say, the factor-level information in Section 3.2 is not well explained.

4. The authors provide a comprehensive ablation study on variants of the proposed methods and show the same trend over different datasets, which is good. However, the codes of the proposed methods are missing, although the authors have ticked on that option in the checklist.

5. Comparing the GraphCL model and other baselines the improvement of the proposed model is impressive. I am wondering whether this approach can be applied to the node-level representation tasks.



**Time Spent Reviewing:**

8

---

> ### Author Response · Authors · 2021-08-10
> **Response to Reviewer ubo5**
>
> We thank the reviewer for the valuable positive feedback. We addressed all the comments. Please kindly find the detailed responses below.
>
> + *Q1. The factor-level information in Section 3.2 is not well explained. It will be good if the authors can do some case studies to show the disentanglement on the learned latent factors.*
>
> **Response:** Thank you for the comment. The formation of a graph is typically driven by many complex latent factors. But for the real-world datasets used in the experiments, such information is not available and thus we cannot explain what our learned latent factors represent. Notice that this is a common case in the disentanglement literature [1] [2].
> To remedy this issue, we conduct experiments on the synthetic graph dataset (Section 4.3). The synthetic graphs are generated from eight factors (i.e., probability p that two nodes are connected in a community). We observe that our method achieves the best results when the number of channels K is equal to the ground-truth number of latent factors, indicating we can model the underlying structure of this simulation dataset. Figure 4 also shows that the latent factors learned by our method, where each block represents one factor. The block-wise correlation pattern indicates that the eight channels of the disentangled graph encoder in our method are able to capture mutually exclusive information and the latent features have indeed been disentangled.
>
> [1] Disentangled Graph Convolutional Networks. ICML 2019.
> [2] Factorizable Graph Convolutional Networks. NeurIPS 2020.
>
> + *Q2. The codes of the proposed methods are missing.*
>
> **Response:** Thank you for the comment. We have provided the detailed generating method of the synthetic datasets, the URLs of real-world datasets, pseudo-code and experiments settings in the supplementary material. We promise to release the source code at publication time.
>
> + *Q3. I am wondering whether this approach can be applied to the node-level representation tasks.*
>
> **Response:** Thank you for the question. In this work, we focus on the disentangle graph contrastive learning for graph-level representations. The disentangled graph encoder is tailored to capture multiple aspects of graphs through learning disentangled latent factors of graphs and thus cannot be directly applied to the node-level representation tasks. Nevertheless, we think a similar methodology of our method could be extended to node-level tasks if we can design a new encoder to infer the latent factors of each node and discrimination tasks of each latent factor of the nodes.

---

> ### Author Response · Authors · 2021-08-29
> **Kind reminder to the reviewer**
>
> Dear Reviewer,
>
> We are wondering whether your concerns have been properly addressed.
>
> If you have further questions after reading the responses, it would be great to let us know.
>
> Best regards,
>
> The Authors

---

### Official Review · Reviewer_9Pj4 · 2021-07-15

**Rating:** 7
**Confidence:** 3

**Summary:**

The submission presents Disentangled Graph Contrastive Learning (DGCL), a method to learn disentangled representations for graphs. To this end, the submission proposes a multi-channel message passing graph encoder. Each channel is supposed to match one disentangled latent factor. Furthermore, contrastive learning is performed on each latent factor separately. The experiments show that the learned representations work better than several prior works in several datasets when used as input for an SVM.

**Main Review:**

Overall, I found the method proposed by the paper interesting and idea for a disentangled representation of graphs to be novel.

A main issue in the evaluation is that an SVM is used to perform the final task prediction based on the learned graph embeddings. SVMs are a specific kind of model and representations that work well with SVMs might not work well with other models such as neural networks. Hence, I think the paper could be substantially improved if more models are used to perform prediction tasks based on the learned representations. The paper argues that using SVMs is a standard approach, which is also used in prior works. However, this does not mitigate the fact that the evaluation is limited.

Furthermore, fine-tuning with a small set of labeled graphs could have been performed to better evaluate the learned representations. In fact, this would perhaps be the most prominent use case for the learned representations in analogy to other self-supervised representations, e.g. for natural language. Furthermore, successful fine-tuning would be important to evaluate the significance of the results of the submission.

Another issue is the fact that the learned latent factors in the real world datasets are not analyzed beyond their performance. It would have been very interesting to see 1) what the learned latent factors represent, and 2) if they are really mutually exclusive. This analysis is only performed for the synthetic data. I wonder if the latent factors in the real datasets are similarly mutually exclusive.

I would have preferred to also see results of supervised methods, e.g. in Table 1, to better understand if self-supervised learning alone is already able to learn reasonable good representations.

In general, I found the paper easy to read and easy to understand, even though a rather complex/technical method is proposed. However, I didn't find the contribution bullet points (lines 77-87) to be helpful and rather repetitive. Some minor paper also also slightly repetitive, e.g. Cosine Similarity is introduced in line 112 and then again in line 165.

One part that I did not really understand is why the latent factors are mutually exclusive. The paper states this multiple times (e.g. in line 218), but I couldn't identify a single part in the approach that forces the latent factors to be mutually exclusive. Perhaps I missed something here.

An interesting evaluation on the number of channels is provided in the supplementary material. I found this evaluation very helpful to understand if and how multiple channels improve the representations. I think it would improve the paper if this analysis could be moved into the main part of the paper, since the idea to use multiple disentangled channels is the key contribution of the submission.

**Time Spent Reviewing:**

2.5

---

> ### Author Response · Authors · 2021-08-10
> **Response to Reviewer 9Pj4**
>
> We thank the reviewer for the insightful positive comments and acknowledgment of our novelty. Please kindly find the detailed responses to the comments below.
>
> + *Q1. More models should be used to perform prediction tasks based on the learned representations. The paper argues that using SVMs is a standard approach, which is also used in prior works. However, this does not mitigate the fact that the evaluation is limited.*
>
> **Response:** Thank you for this comment. Following your suggestion, we have added experiments that use neural network (one fully-connected layer) to perform the final prediction based on the learned graph embeddings. The experimental results are shown below. We observe that the disentangled graph representations learned by our method also work well with the neural network classifier, besides the SVM classifier.
>
> |                  | **MUTAG**  | **PTC-MR** | **PROTEINS** |  **NCI1**  | **IMDB-B** | **IMDB-M** | **RDT-B**  | **RDT-M5K** | **COLLAB** |
> | :------------: | :--------: | :--------: | :----------: | :--------: | :--------: | :--------: | :--------: | :---------: | :--------: |
> |      SVM       | 92.1 ± 0.2 | 65.8 ± 1.5 |  76.4 ± 0.5  | 81.9 ± 0.2 | 75.9 ± 0.7 | 51.9 ± 0.4 | 92.7 ± 0.2 | 56.1 ± 0.2  | 81.2 ± 0.3 |
> | Neural Network | 91.8 ± 0.3 | 67.1 ± 1.3 |  75.1 ± 0.6  | 79.1 ± 0.5 | 76.8 ± 0.6 | 49.9 ± 0.3 | 92.8 ± 0.3 | 55.8 ± 0.3  | 78.4 ± 0.5 |
>
> + *Q2. Successful fine-tuning would be important to evaluate the significance of the results of the submission.*
>
> **Response:** Following your suggestion, we have added experiments that perform fine-tuning.  The fine-tuning requires a small number of labeled graphs to optimize the graph encoder after pretraining. The rate of labeled graphs for fine-tuning is set to 10%. We compare our method with two strong self-supervised methods, InfoGraph and GraphCL. The results are shown as follows. We can observe that our proposed method achieves strong performance under the fine-tuning setting.
>
> |           |  PROTEINS  |   IMDB-B   |   RDT-B    |  RDT-M5K   |
> | :-------: | :--------: | :--------: | :--------: | :--------: |
> | InfoGraph | 72.3 ± 0.4 | 71.3 ± 0.8 | 88.7 ± 1.0 | 53.6 ± 0.3 |
> |  GraphCL  | 74.2 ± 0.3 | 70.7 ± 0.4 | 89.1 ± 0.2 | 52.6 ± 0.5 |
> |   **DGCL**    | 74.3 ± 0.2 | 72.5 ± 0.5 | 89.3 ± 0.3 | 52.5 ± 0.8 |
>
> + *Q3. It would have been very interesting to see 1) what the learned latent factors represent, and 2) if they are really mutually exclusive. This analysis is only performed for the synthetic data. I wonder if the latent factors in the real datasets are similarly mutually exclusive.*
>
> **Response:** Thank you for the comment. The formation of a graph is typically driven by many complex latent factors. But for the real-world datasets used in the experiments, such information is not available and thus we cannot explain what our learned latent factors represent. Notice that this is a common case in the disentanglement literature [1] [2]. To remedy this issue, we conduct experiments on the synthetic graph dataset (Section 4.3). The synthetic graphs are generated from eight factors (i.e., probability p that two nodes are connected in a community). We observe that our method achieves the best results when the number of channels K is equal to the ground-truth number of latent factors, indicating we can model the underlying structure of this simulation dataset. Figure 4 also shows that the latent factors learned by our method, where each block represents one factor. The block-wise correlation pattern indicates that the eight channels of the disentangled graph encoder in our method are able to capture mutually exclusive information and the latent features have indeed been disentangled.
>
> Following your suggestion, we have conducted the analysis of latent factors for real-world graph datasets and observe similar patterns, i.e., our learned latent factors for the real-world datasets are mutually exclusive. We will add this analysis in the revised version.
>
> [1] Disentangled Graph Convolutional Networks. ICML 2019.
> [2] Factorizable Graph Convolutional Networks. NeurIPS 2020.
>
> + *Q4. I would have preferred to also see results of supervised methods, e.g., in Table 1, to better understand if self-supervised learning alone is already able to learn reasonable good representations.*
>
> **Response:** Thank you for this suggestion. We have obtained the results (see the following table) of some popular supervised methods and will add them into Table 1 in the revised version of this paper as you suggest. We can observe similar phenomenon with [1], [2] that self-supervised learning is able to learn good graph representations, achieving comparable results with supervised methods. As one type of self-supervised graph representation learning methods, our proposed method can learn the disentangled representations and achieve the best performance compared with other self-supervised baselines and classical unsupervised methods.
>
> |          | **MUTAG**  | **PTC-MR** | **PROTEINS** |  **NCI1**  | **IMDB-B** | **IMDB-M** | **RDT-B**  | **RDT-M5K** | **COLLAB** |
> | :------: | :--------: | :--------: | :----------: | :--------: | :--------: | :--------: | :--------: | :---------: | :--------: |
> |   GCN    | 85.6 ± 5.8 | 64.2 ± 4.3 |  75.7 ± 2.0  | 64.4 ± 1.5 | 74.0 ± 3.4 | 51.9 ± 3.8 | 50.0 ± 0.0 | 20.0 ± 0.0  | 79.0 ± 1.8 |
> |   GAT    | 89.4 ± 6.1 | 66.7 ± 5.1 |  65.7 ± 2.3  | 58.5 ± 1.1 | 70.5 ± 2.3 | 47.8 ± 3.1 | 85.2 ± 3.3 | 21.1 ± 0.2  | 53.6 ± 2.5 |
> |   GIN    | 89.4 ± 5.6 | 64.6 ± 7.0 |  76.2 ± 2.8  | 82.7 ± 1.7 | 75.1 ± 5.1 | 52.3 ± 2.8 | 92.4 ± 2.5 | 57.5 ± 1.5  | 80.2 ± 1.9 |
> | **DGCL** | 92.1 ± 0.2 | 65.8 ± 1.5 |  76.4 ± 0.5  | 81.9 ± 0.2 | 75.9 ± 0.7 | 51.9 ± 0.4 | 92.7 ± 0.2 | 56.1 ± 0.2  | 81.2 ± 0.3 |
>
> [1] Contrastive Multi-View Representation Learning on Graphs. ICML 2020.
> [2] GCC: Graph Contrastive Coding for Graph Neural Network Pre-Training. KDD 2020.
>
> + *Q5. The paper is easy to read and easy to understand, even though a rather complex/technical method is proposed. However, there exist some repetitive writing including the contribution bullet points (lines 77-87), Cosine Similarity line 112 and line 165.*
>
> **Response:** Thank you for the positive comments. We will also carefully improve the presentation of contributions in bullet points (lines 77-87) and remove the repetitive description of cosine similarity in line 165 in the revised version of the paper as you suggest.
>
> + *Q6. I couldn't identify a single part in the approach that forces the latent factors to be mutually exclusive. Perhaps I missed something here.*
>
> **Response:** Thank you for the comment. We have clarified how to force the latent factors to be mutually exclusive (disentangled) as follows (Section 3.4, line 218-225 in the paper). (1) The multi-channel disentangled graph encoder can summarize the specific aspect of the graph according to the corresponding latent factor for each channel. (2) We factorize the instance discrimination into K factor-wise subtasks that will enforce the disentanglement of the learned graph representations. (3) The inference of latent factors is computed based on the k-th (see the numerator in Eq. (6)) and other K−1 latent factors (see the denominator in Eq. (6)). As a result, the graph encoder is forced to preserve exclusive information in each channel to get more accurate approximation to the posterior, if a tighter ELBO is expected. The strong inductive biases encourage to learn disentangled graph representations that match the ground truth factors behind the graphs.
>
> + *Q7. I think it would improve the paper if this analysis (evaluation on the number of channels) could be moved into the main part of the paper.*
>
> **Response:** Thank you for this kind suggestion. We will try to reorganize and move this analysis into the main paper, if page limits allow.

---

> ### Author Response · Authors · 2021-08-29
> **Kind reminder to the reviewer**
>
> Dear Reviewer,
>
> We are wondering whether your concerns have been properly addressed.
>
> If you have further questions after reading the responses, it would be great to let us know.
>
> Best regards,
>
> The Authors

---

### Official Review · Reviewer_mcrT · 2021-07-16

**Rating:** 6
**Confidence:** 4

**Summary:**

This manuscript proposes a disentangled graph contrastive learning method, which can generate the graph-level disentangled representation. The proposed method contains two main parts: the graph encoder to generate the factors and the factor-wise contrastive learning strategy to learn the parameters of the encoder.


**Limitations And Societal Impact:**

Please see the problems in the main review

**Main Review:**

Pros:
1, This manuscript tries to solve the disentangled graph learning in an unsupervised manner.
2, The proposed method is evaluated on several datasets and indeed shows its effectiveness.

Some problems:
1, For a fair comparison, it would be better to also show the number of parameters and the inference time for the proposed method and the comparisons.

2, The work “Factorizable graph convolutional networks” utilizes different GNNs on different factor graphs to generate the factor features, which is similar to the proposed disentangled graph encoder. It is recommended to discuss the connection and difference between these two methods.

3, It seems that the proposed optimization method (disentangled factor-wise contrastive learning) is a general method that can not only be applied to the graph contrastive learning problem (once we can obtain the contrastive pair examples). I’m wondering if there is any special reason to only apply the proposed method to the graph learning problem?


Minor issues:
1, It is recommended to use the same font in the figures like that in the main body, like Fig.2 and Fig.3.

**Time Spent Reviewing:**

2 hours

---

> ### Author Response · Authors · 2021-08-10
> **Response to Reviewer mcrT**
>
> We thank the reviewer for the insightful positive comments. Please kindly find the detailed responses to the comments below.
>
> + *Q1.1. It would be better to show the number of parameters for the proposed method and the comparisons.*
>
> **Response:** Thank you for this comment. We add the comparisons as follows. For our method, the number of parameters is O(L*K*(d/K)^2+K*(d/K)), i.e., O(Ld^2/K), where K is the number of channels, L is the number of message-passing layers of disentangled graph encoder, and d is the dimensionality of the embeddings. Specifically, because we adopt GIN as the message passing layers, the number of parameters of the disentangled graph encoder is O(L*K*(d/K)^2). The number of parameters of K latent factor prototypes is O(K*(d/K)). For the baselines that using Graph Neural Network (i.e., GCN or GIN) as the graph encoder, including GVAE, InfoGraph, GCC, MVGRL, and GraphCL, the number of parameters is O(Ld^2). Therefore, the number of parameters of the proposed method and the baselines are comparable. If let L=2, K=2, d=64, the number of parameters of the proposed method and one self-supervised baseline GraphCL is about 12,000, 13,000 respectively. We will add the analysis above in the revised version.
>
> + *Q1.2. It would be better to also show the inference time for the proposed method and the comparisons.*
>
> **Response:** Thank you for this suggestion. We analyze the time complexity of our proposed method DGCL and baselines as follows. The time complexity of training and inference of our method is O(M), where M denotes the number of edges in the graphs. The self-supervised baselines (e.g., GraphCL, MVGRL, etc.,) also have a time complexity of O(M). Because VAE-based methods require the reconstruction of adjacency matrix, the training time complexity is O(N^2), where N denotes the number of nodes in the graphs. In conclusion, our method has lower complexity than those VAE-based methods and comparable complexity with self-supervised baselines.
>
> We have also conducted experiments to validate the efficiency. For example, on COLLAB dataset, it takes around 89.75s to train our DGCL and 71.63s to train GraphCL (one self-supervised baseline) for 20 epochs (with the same hyperparameters) respectively. However, it takes 310.27s to train GVAE (one unsupervised baseline) for 20 epochs. The analysis above can demonstrate the efficiency of our method, and we will add it into the revised paper as you suggest.
>
> + *Q2. It is recommended to discuss the connection and difference between “Factorizable graph convolutional networks” and this work.*
>
> **Response:** Thank you for this suggestion. We discuss the connection and difference as follows. (1) Connection: Both methods focus on the important and novel problem of how to learn disentangled graph-level representations. (2) Differences: The main difference lies in that our method studies disentangled self-supervised graph representation learning while FactorGCN is a supervised disentanglement method. Therefore, FactorGCN requires enough labels to train the model. On the contrary, our method can learn disentangled graph representations effectively without task-dependent annotated labels which are extremely scarce, or even unavailable in practice. We will add these discussions in the revised version of this paper.
>
> + *Q3. It seems that the proposed optimization method (disentangled factor-wise contrastive learning) is a general method that can not only be applied to the graph contrastive learning problem (once we can obtain the contrastive pair examples). I’m wondering if there is any special reason to only apply the proposed method to the graph learning problem?*
>
> **Response:** Thank you for this comment. We think that disentangled self-supervised graph representation learning is novel and important so we study this problem for the first time starting from the field of graphs. We design a tailored disentangled graph encoder so that it can be sufficiently expressive to infer the disentangled latent factors in the graph. The disentangled graph encoder is specific to the field of graphs. We also introduce the factor-wise contrastive learning approach on tailored discrimination tasks in terms of each latent factor independently. This method may be extended to other fields, which is a promising direction to the future work.
>
> + *Q4. It is recommended to use the same font in the figures like that in the main body, like Fig.2 and Fig.3.*
>
> **Response:** Thank you for this suggestion. We will adopt the same font in Fig.2 and Fig.3 like that in the main body in the revised version of this paper.

---

> ### Author Response · Authors · 2021-08-29
> **Kind reminder to the reviewer**
>
> Dear Reviewer,
>
> We are wondering whether your concerns have been properly addressed.
>
> If you have further questions after reading the responses, it would be great to let us know.
>
> Best regards,
>
> The Authors

---

### Official Review · Reviewer_vaM6 · 2021-07-19

**Rating:** 5
**Confidence:** 4

**Summary:**

This paper proposes a graph classification model named Disentangled Graph Contrastive Learning (DGCL). The core idea of the paper is to learn a disentangled graph-level representation for each graph in a contrastive self-supervised learning manner. Extensive experiments demonstrate the effectiveness and superiority of the proposed model over the baselines in the paper.

**Limitations And Societal Impact:**

Limitations have been discussed in the supplementary materials.

**Main Review:**

The strong points of the paper are as follows:

S1. Current self-supervised learning methods for GNNs neglect the entanglement of the latent factors and thus result in the suboptimal performance for downstream tasks. In the paper, the authors propose to learn disentangled graph-level representations with self-supervision. Then the disentangled representation can be used to boost the performance of the downstream tasks.

S2. The proposed factor-wise contrastive learning approach guarantees the ability of the factorized representations to reﬂect the expressive information from different latent factors independently.

S3. The paper is written and organized in a good manner.

Nevertheless, there are also several weak points:

W1. The novelty of this paper is relatively marginal with insignificant technical contribution since the proposed method is mostly a combination of existing modules. The key challenges to the problem are weakly discussed (without sound theoretical or empirical analysis), rendering the necessity of considering disentangled SSL unconvincing.

W2. Unsupervised ablation of GIN should be included in the baselines, since DGCL applies GIN as the backbone model. Also, DGI needs to be compared.

W3. The efficiency and scalability of the proposed model and baselines are not discussed in the paper.


**Time Spent Reviewing:**

5

---

> ### Author Response · Authors · 2021-08-10
> **Response to Reviewer vaM6**
>
> We thank the reviewer for the valuable feedback. Please kindly find the detailed responses to the comments below.
>
> + *Q1. The novelty of this paper is relatively marginal. The key challenges to the problem are weakly discussed.*
>
> **Response:** We would like to clarify the novelty of this work. In literature, existing self-supervised graph representation learning methods generally characterize graphs as a perceptual whole, failing to capture different aspects of the graphs in terms of latent factors. Yet the formation of a real-world graph is typically driven by many complex latent factors. In this work, our proposed method is able to capture different aspects of the graphs through learning disentangled graph representations with self-supervised learning. The latent factors can be captured by the multi-channel graph encoder and the graph representations are encouraged to be disentangled by the factor-wise contrastive algorithm.
>
> More discussions on the key challenges of this problem are as follows. (1) Given that the latent factors driving the formation of a graph are often complex and unobserved, it will be challenging to identify these complex latent factors behind the graphs. (2) To date, it still remains unclear how we can design a tailored graph encoder which is sufficiently expressive to capture multiple aspects of graphs. (3) It is also challenging to design discrimination tasks in terms of each latent factor independently to encourage the disentanglement. (4) The optimization for the objective function is difficult due to the latent factors, making it challenging to derive the approximate objective function (e.g., ELBO) to optimize the parameters.
>
>
> + *Q2.1. Unsupervised ablation of GIN should be included in the baselines, since DGCL applies GIN as the backbone model.*
>
> **Response:** Thank you for this comment. In our experiments, we have already compared with some self-supervised baselines using GIN as the graph encoder. To further compare GIN with unsupervised objectives as you suggested, we adopt three objective functions to train GIN: variational graph auto-encoders [1], Beta-VAE [2], and FactorVAE [3].  The experimental results are shown below. We observe that DGCL consistently achieves the best performance compared with these additional unsupervised variants of GIN.
>
> |                | **MUTAG**  | **PTC-MR** | **PROTEINS** |  **NCI1**  | **IMDB-B** | **IMDB-M** | **RDT-B**  | **RDT-M5K** | **COLLAB** |
> | :------------: | :--------: | :--------: | :----------: | :--------: | :--------: | :--------: | :--------: | :---------: | :--------: |
> |      GVAE      | 87.7 ± 0.7 | 61.2 ±1.8  |  67.9 ± 0.7  | 53.7 ± 0.5 | 70.7 ± 0.7 | 49.3 ± 0.4 | 87.1 ± 0.1 | 52.8 ± 0.2  | 55.6 ± 3.1 |
> |    GBetaVAE    | 87.9 ± 0.7 | 62.0 ± 1.9 |  73.4 ± 0.4  | 75.7 ± 1.2 | 70.8 ± 0.4 | 49.4 ± 0.7 | 87.3 ± 0.4 | 53.1 ± 0.6  | 68.1 ± 2.1 |
> |   GFactorVAE   | 88.6 ± 0.6 | 62.2 ± 1.6 |  74.9 ± 0.6  | 71.8 ± 2.1 | 71.1 ± 0.5 | 49.5 ± 0.6 | 87.5 ± 0.5 | 53.8 ± 0.7  | 68.4 ± 2.2 |
> | **DGCL(Ours)** | 92.1 ± 0.2 | 65.8 ± 1.5 |  76.4 ± 0.5  | 81.9 ± 0.2 | 75.9 ± 0.7 | 51.9 ± 0.4 | 92.7 ± 0.2 | 56.1 ± 0.2  | 81.2 ± 0.3 |
>
> [1] Variational graph auto-encoders. NeurIPS 2016.
> [2] beta-VAE: Learning Basic Visual Concepts with a Constrained Variational Framework. ICLR 2017.
> [3] Disentangling by Factorising. ICML 2018.
>
> + *Q2.2. Also, DGI needs to be compared.*
>
> **Response:** Thank you for the comment. We focus on the graph-level representation learning in this work. DGI can only learn node-level representations and thus cannot be applied to our tasks. InfoGraph is an extension of DGI to learn graph-level representations and we have adopted InfoGraph as one of our baselines.
>
> + *Q3. The efficiency and scalability of the proposed model and baselines are not discussed in the paper.*
>
> **Response:** Thank you for this suggestion. We analyze the efficiency and scalability of the proposed method as follows. The time complexity of our method DGCL is O(M), where M denotes the number of edges in the graphs. Specifically, DGCL adopts GIN as the message-passing layers so the time complexity of the disentangled graph encoder is O(M). As for the factor-wise contrastive learning, the positive and negative samples are drawn from graph data augmentations and graphs from the same minibatch respectively, which will not induce a higher computational cost. The time complexity of self-supervised learning baselines (e.g., GraphCL, MVGRL, etc.) in the experiments is also O(M). Therefore, the time complexity of our proposed DGCL is on par with these baselines. Notice that GVAE, one of the unsupervised baselines, has O(N^2) time complexity due to the adjacency matrix reconstruction in the VAE framework, where N denotes the number of nodes in the graphs. Since O(M) << O(N^2) for sparse graphs, our proposed method is much more scalable than GVAE.
>
> For the efficiency aspect, the training time of our DGCL and two self-supervised baselines (MVGRL, GraphCL) are comparable in practice. On COLLAB, one of the social network datasets in the experiments, it takes around 89.75s to train our DGCL, 175.24s to train MVGRL, and 71.63s to train GraphCL for 20 epochs (with the same hyperparameters) respectively. On NCI1, one of the bioinformatics datasets, it takes around 162.57s to train our DGCL, 203.86s to train MVGRL, and 147.02s to train GraphCL for 20 epochs respectively. The experiments are run on a single GPU (NVIDIA GeForce GTX TITAN X). We will add the analysis above into the revised paper.

---

> ### Author Response · Authors · 2021-08-29
> **Kind reminder to the reviewer**
>
> Dear Reviewer,
>
> We are wondering whether your concerns have been properly addressed.
>
> If you have further questions after reading the responses, it would be great to let us know.
>
> Best regards,
>
> The Authors

---

> ### Author Response · Authors · 2021-09-02
> **Response to Reviewer vaM6**
>
> Dear Reviewer vaM6,
>
> We thank you again for the valuable and detailed comments.
>
> We hope that we have adequately addressed your concerns on the clarification of novelty, unsupervised ablation of GIN, and the efficiency and scalability of the proposed method. We deeply appreciate your feedbacks that help to further improve our work, please kindly let us know if there is anything unclear from your side.

---

### Decision · Program_Chairs · 2021-09-28

**Decision:**

Accept (Poster)

**Comment:**

The manuscript has been reviewed by four experienced reviewers, among whom three voted for acceptance and one reviewer (vaM6) voted for borderline reject. The borderline reviewer mainly complained about novelty, some missing experiments, as well as the discussion on scalability and efficiency. Regarding the novelty, in fact, other reviewers indeed rated the proposed approach as being novel; regarding the missing experiments, the authors claimed that the requested experiments were already in the manuscript; regarding the scalability and efficiency, the authors provided a discussion in the rebuttal.

Since reviewer vaM6 only mildly rejects the submission while all other reviewers tend to accept, the AC will follow the majority vote and accept the submission. Still, the authors are strongly recommended to account for the comments from all reviewers in the revised version.



**Consistency Experiment:**

NeurIPS has a long history of experimentation. In 2014, NeurIPS ran an experiment in which 10% of submissions were reviewed by two independent committees to quantify the randomness in the review process. This year, we repeated a variant of this experiment to see how the quality of the review process has changed over time.  This paper was part of the experiment and was therefore assigned to two committees (consisting of reviewers, an Area Chair, and a Senior Area Chair) that reached independent decisions.  If both committees made the same recommendation, this recommendation was followed. If a single committee recommended acceptance, the paper was accepted (with the exception of a few cases in which the other committee identified what we considered a fatal flaw, e.g., an error in a key result).

Both committees reached the same decision: **Accept (Poster)**

The other committee assigned to the paper recommended **Accept (Poster)**.  You can find the other set of reviews, along with any follow up discussion with the authors here:
https://openreview.net/forum?id=C_L0Xw_Qf8M